# Isolation, Molecular Identification and Mycotoxin Profile of Fusarium Species Isolated from Maize Kernels in Iran

**DOI:** 10.3390/toxins11050297

**Published:** 2019-05-24

**Authors:** Maryam Fallahi, Hossein Saremi, Mohammad Javan-Nikkhah, Stefania Somma, Miriam Haidukowski, Antonio Francesco Logrieco, Antonio Moretti

**Affiliations:** 1Department of Plant Protection, Faculty of Agricultural Science and Engineering, College of Agriculture and Natural Resources, University of Tehran, Karaj 77871-31587, Iran; m.fallahi@ut.ac.ir (M.F.); jnikkhah@ut.ac.ir (M.J.-N.); 2Institute of Sciences of Food Production, Research National Council (ISPA-CNR), via Amendola 122/O, 70126 Bari, Italy; stefania.somma@ispa.cnr.it (S.S.); miriam.haidukowski@ispa.cnr.it (M.H.); antonio.logrieco@ispa.cnr.it (A.F.L.)

**Keywords:** mycotoxin, maize ear rot, fumonisin, trichothecenes, toxigenic risk

## Abstract

*Fusarium* species are among the most important fungal pathogens of maize, where they cause severe reduction of yield and accumulation of a wide range of harmful mycotoxins in the kernels. In order to identify the *Fusarium* species and their mycotoxin profiles associated to maize ear rot and kernel contamination in Iran, a wide sampling was carried out from field in ten major maize-producing provinces in Iran, during 2015 and 2016. From 182 samples of maize kernels, 551 strains were isolated and identified as belonging to *Fusarium* genus. Among the 234 representative strains identified at species level by translation elongation factor (*EF-1α*) sequences, the main *Fusarium* species were *F. verticillioides* and *F. proliferatum*, together representing 90% of the Iranian *Fusarium* population, and, to a lesser extent, *F. incarnatum equiseti* species complex (FIESC)*, F. thapsinum* and *F. redolens*. Fumonisin (FBs) production by *F. verticillioides* and *F. proliferatum* representative strains was analysed, showing that all strains produced FB_1_. None of *F. verticillioides* strains produced FB_2_ nor FB_3_, while both FB_2_ and FB_3_ were produced only by *F. proliferatum*. Total mean of FBs production by *F. verticillioides* was higher than *F. proliferatum*. The occurrence of different *Fusarium* species on Iranian maize is reason of great concern because of the toxigenic risk associated to these species. Moreover, the diversity of the species identified increases the toxigenic risk associated to *Fusarium* contaminated maize kernels, because of the high possibility that a multi-toxin contamination can occur with harmful consequences on human and animal health.

## 1. Introduction

Maize (*Zea mays* L.) is one of the most important cereals in the world for animal feed and human use, being the third highest produced grain crop, after wheat and rice. About 65% of the total world maize production is used as livestock feed, 15% as human food, and the remaining 20% is mainly addressed to industrial purposes [1]. However, maize is constantly contaminated by toxigenic fungi (TF), which represent a danger for the potential accumulation of the mycotoxins (MTX) that they produce in the grains at the harvest, and could cause a decrease of the storage life of the final products. Moreover, during the post-harvest period, inappropriate environmental conditions of storage could trigger the further development of TF in the grains with consequent production of MTX and further deterioration of maize kernels with low quality and nutritive value [2]. On the other hand, in the field, TF ability to colonise ears of maize is strictly related to climatic parameters, which are also key factors for the MTX production within the kernels. The MTX that cause main concern regarding maize, because of their toxicity and their worldwide distribution on the most common crops, are aflatoxins (AFLA), fumonisins (FBs), ochratoxin A (OTA), trichothecenes and zearalenone (ZEA) [3,4], produced by a wide range of species belonging to three fungal genera, *Aspergillus*, *Fusarium* and *Penicillium*.

In particular, maize ear rot is among the most serious diseases and is caused by a broad range of *Fusarium* species that significantly reduce the quantity and quality of maize. Commonly, this disease is divided in two types, pink ear rot and red ear rot. Pink ear rot is caused by species belonging to the *Fusarium fujikuroi* species complex (FFSC) [5] that include species that can colonise temperate and tropical regions and are often regarded as endophytes. On the other hand, red ear rot is caused mostly by *F. graminearum* and related species such as *F. culmorum*, and is more developed in the geographical areas where colder temperatures and more humid conditions characterise the environment [6]. In Iran, most of the reports have shown that the species belonging to FFSC were the most common in maize.

Among the FFSC, *F. verticillioides* and *F. proliferatum* are those that cause major concern, since they are both the main producing species of the carcinogenic FBs [7] and very common in maize in all regions of the world where this crop is cultivated, included Iran. As a consequence of such colonisation, FBs’ contamination in maize kernels in the fields at maturity and, as carry over, in the silos during storage, is often high [8,9]. Fumonisins have been related to several animal and human diseases such as esophageal cancer (EC), as reported in several countries worldwide [10]. In particular, in Iran, several studies have reported not only high levels of FBs contamination and related *Fusarium* species occurrence in Iranian maize and maize-based products [11,12,13,14], but also a positive correlation between FBs content in maize kernels and the risk of EC [9]. More detailed studies on the fungal mycoflora associated with maize kernels have also revealed in some maize-producing areas of Iran that *F. verticillioides* and *F. proliferatum* were the prevalent species [15,16]. 

On the other hand, more recently, Amirahmadi et al. [17] reported the occurrence of deoxynivalenol (DON), T-2 toxin and ZEA that are mycotoxins produced by other *Fusarium* species than *F. verticillioides* and *F. proliferatum*. Deoxynivalenol and T-2 are the most important members of trichotecene B and A groups, respectively, and are considered potent inhibitors of proteic synthesis, while ZEA is a metabolite causing estrogenic disorders in human and animals [10]. However, all these reports were limited for the geographical areas investigated in Iran and could not provide a complete overview of the situation in the country. This is extremely important, since a more comprehensive knowledge of the *Fusarium* species profile occurring on maize kernels, would offer a more reliable assessment of the related mycotoxin risk, since each *Fusarium* species has its own mycotoxin profile [10]. 

Therefore, since maize is the most important cereal crop in Iran, and its consumption is related to high risk for the population due to possible *Fusarium* mycotoxin contamination, there is a need to correctly assess the main *Fusarium* species occurring on maize in the country, and detect their mycotoxin profile. The aims of this study were: (i) to significantly sample maize kernels at maturity in the all main producing regions in Iran; (ii) to identify the *Fusarium* species occurring in the samples and (iii) to assess the mycotoxin profiles of the main *Fusarium* species identified.

## 2. Results

### 2.1. Species Identification

In total, 551 *Fusarium* isolates were obtained from maize samples and, based on morphological characteristics, several different species were identified, namely *F. verticillioides*, *F. proliferatum*, *F. subglutinans*, *F. thapsinum*, *F. redolens*, *F. sacchari* and species belonging to *F. solani* species complex (FSSC) and *F. incarnatum equiseti* species complex (FIESC). The number of maize samples and the number of the isolates obtained from the ten major maize-producing Iranian provinces are reported in Table 1. 

To correctly assign the *Fusarium* species identification, 234 isolates, selected to be representative of the *Fusarium* population isolated from Iranian maize, were molecularly identified by DNA sequence analysis of *EF-1α* gene. The Figure 1 shows the phylogenetic tree, based on *EF-1α* gene, inferred for 234 sequences compared to *Fusarium* species reference sequences available on GeneBank. This analysis allowed to identify the isolates as belonging to *F. verticillioides, F. proliferatum, F. subglutinans, F. thapsinum, F. temperatum, F.nygamai,* FIESC, *F. brachygibbosum, F.redolens* and FSSC. In particular, 145 isolates were identified as *F. verticillioides*, 46 as *F. proliferatum*, 10 *F. thapsinum*, 3 *F. subglutinans*, 1 *F. temperatum*, 1 *F. nygamai*, 5 *F. redolens*, 1 isolate belonging to FSSC, 1 *F. brachygibbosum* and 21 isolates as belonging to FIESC. All isolates described as *F. sacchari*, based on morphological characteristics, were identified as *F. subglutinans* through molecular analyses.

In Figure 2, the distribution of the mainly occurring *Fusarium* species from Iranian maize has been reported, with the number of strains identified for each species. An overview of the occurrence of the most present species, i.e., *F. verticillioides* and *F. proliferatum*, expressed as percentage of these species on total *Fusarium* species, is shown in Table 1, referred to the considered Iranian provinces. Moreover, based on the samples collected from each province, the incidence of *Fusarium* species and the incidence of both *F. verticillioides* and *F. proliferatum* were calculated as percentage of seeds infected on total analysed seeds, and reported in Table 1.

Incidence of 15% of seeds infected by *Fusarium* species was observed in Iranian maize, ranging between 33% in Golestan province and 5% in Esfahan. In Golestan was also observed the highest incidence of *F. verticillioides* (20%), while the lowest incidence of this species was reported in Lorestan. About *F. proliferatum* incidence (5%), the highest values were observed in Alborz (10%) and Golestan (9%), while this species was absent in Khuzestan and in Esfahan. In all the Iranian provinces, maize kernels were more contaminated by *F. verticillioides* than *F. proliferatum*, except in Alborz, in which the incidence of both the species was the same (10%). 

Among the 551 *Fusarium* strains isolated from maize kernels, *F. verticillioides* species occurred at 59% and *F. proliferatum* species at 31%, representing together 90% of the Iranian *Fusarium* population. In most of the Iranian provinces the percentage of *F. verticillioides* was higher than *F. proliferatum*, and above 50%, except in Lorestan, where *F. proliferatum* represents the most of the *Fusarium* population (55%) and in Alborz, where both the species occurred at 47 and 48%, respectively (Table 1). Other *Fusarium* species were isolated from Iranian maize kernels (10%), mostly FIESC (6%), with higher rate in Golestan, Fars, Khuzestan and Lorestan (11, 16, 26, 27%, respectively).

### 2.2. Mycotoxin Production

Fumonisin production was assessed for all the analysed strains. In particular, as reported in Table 2, all *F. verticillioides* (67) and *F. proliferatum* (26) strains produced FB_1_, with mean values of 505 and 216 µg/g, respectively. All *F. verticillioides* strains produced FB_1_ levels above 150 and up to 2232 µg/g, except two strains (about 80 µg/g); but none of them produced detectable levels of FB_2_ or FB_3_. Conversely, FB_1_ production of *F. proliferatum* strains was more variable, with values ranging from 1 to 1860 µg/g and a mean of 216 µg/g; in addition, 17 *F. proliferatum* strains on 26 (65%) also produced FB_2_ and 11 strains (42%) produced FB_3_, with mean values of 30 and 3 µg/g, respectively. In particular, FB_2_ production was never higher than 25 µg/g, except for one strain (ITEM 18177) that produced 466 µg/g. Total FBs produced by *F. proliferatum* ranged between 1 to 2335 µg/g, with mean value of 238 µg/g. In addition to *F. verticillioides* and *F. proliferatum*, the only *F. nygamai* strain (ITEM 18217) was also analysed for FBs production, showing to be able to produce FB_1_ (38 µg/g).

Three isolates of *F. subglutinans*, one isolate of *F. temperatum* and 5 of *F. redolens* were tested for production of enniatins (ENNs) A, A_1_, B, B_1_ and beauvericin (BEA). None of them produced ENN A, A_1_ and B; while only two *F. redolens* strains out of 5 (ITEM 18207 and ITEM 18208) produced ENN B_1_, both at values of 124 µg/g.

Production of BEA was detected for one *F. subglutinans* strain out of 3 (ITEM 18269, 16 µg/g), only for *F. temperatum* strain ITEM 18272 (302 µg/g), and for 4 *F. redolens* strains out of 5 (ITEM 18207, ITEM 18208, ITEM 18265 and ITEM 18266). In particular these *F. redolens* strains were shown to be able to produce high levels of BEA, ranging between 688 and 7936 µg/g, with mean value of 4414 µg/g. 

For a detailed description of MTX production of each strain, a Appendix A has been provided.

## 3. Discussion

This study represents the first extended monitoring on the *Fusarium* species colonising maize in Iran regions where maize is the main cultivation. Several reports have described occurrence and incidence of main *Fusarium* species in Iran, but they were mostly all limited in regional sampling, that usually provided only a restricted view of the risk related to *Fusarium* contamination of maize in the country [9,11,12,13,14]. Moreover, most of studies were focused on a single species such as *F. verticillioides* [13], or a single *Fusarium* MTX occurrence [9,18,19]. On the other hand, Rahjoo et al. [16] reported, as far as we are aware, the results of the unique large maize kernel sampling in Iran, performed from 11 provinces, in 2004 and 2005. In the mentioned paper, *F. verticillioides* and, to a lesser extent, *F. proliferatum*, were the only two species significantly isolated from the maize kernels. Although the number of samples (90) and number of seeds analysed per sample (2) were extremely limited, the data obtained were reason of great concern since both species are well known FB producing species and often co-occur worldwide on maize, making it more difficult the development of resistance program for breeders. Indeed, our study, that expanded the sampling (190 vs. 90) and the number of seeds analysed per sample (20 vs. 2), confirmed *F. verticillioides* and *F. proliferatum* as the main species contaminating maize kernels, with a variable distribution of the two species among the regions sampled. Alizadeh et al. [9] related the increased incidence of EC in some areas of the Iranian Province of Golestan, to the high occurrence of fumonisin B_1_ (FB_1_) in cereals. In our survey, along with Golestan Province, further provinces Alborz, Qazvin and Fars were highly contaminated by both species, showing that a high risk for human and animal health does exist also in these regions, where possible association between FB_1_ contaminated maize and EC should be also investigated. Moreover, the frequency of *F. proliferatum* is high in most of the areas sampled, being very interesting that, in some of the Provinces, *F. proliferatum* occurrence is higher than *F. verticillioides*. These data show a significant shift in the ratio between the two most important FBs producing species compared the previous reports from Iran [9,11,13,16]. 

Climate changes have a significant impact on stages and rates of toxigenic fungi development and can modify host–resistance and host–pathogen interactions, also deeply influencing the conditions for mycotoxin production that vary for each individual pathogen. Moreover, the new combinations mycotoxins/host plants/geographical areas are arousing the attention of the scientific community and require new diagnostic tools and deeper knowledge of both biology and genetics of toxigenic fungi [20]. Therefore, our data trigger further investigations on the possibility that the shift between *F. verticillioides* and *F. proliferatum* in the regions sampled in Iran could be related to the eventual environmental changes that occurred in the same areas during the last decade. 

On the other hand, since the breeding program, at worldwide level, where and if applied, have been addressed to select more resistant maize genotypes only to the *F. verticillioides* colonisation of maize [21], the possibility that *F. proliferatum* could have taken advantage by a reduced occurrence of such competitive fungal species, can also be considered. Finally, the increased incidence of *F. proliferatum* is of further concern because of the wider ability of this species to produce further mycotoxins than FBs, such as BEA, fusaproliferin and moniliformin [6,22]. At worldwide level, among the MTX produced by *F. proliferatum*, only FBs are regulated for their occurrence in row maize or maize by-products [23,24,25]. However, the possibility that the other above mentioned MTX could naturally occur in the maize together with FBs, when *F. proliferatum* colonises the kernels, increases the risk related to the consumption of contaminated maize because of the unpredictable additives, if not synergistic, effects, due to the co-occurrence of multiple MTX produced by this species in the maize [26]. 

In addition to FBs, the detection of other MTX on maize has been poorly reported in the last decade, in Iran. In this respect, a survey from Iran by Amirahmadi et al. [17] reported the occurrence of the two well-known trichothecenes, DON and T-2 toxin, together with ZEA, in maize kernels collected in Ardabil province and local markets in Teheran, while Karami-Osboo et al. [19] detected DON in maize kernels collected in the Iranian provinces of Golestan and Moquan, and Rashedi et al. [18] detected ZEA in maize collected in Chaharmahal and Bakhtiari Province. In all these studies, no data were provided on the *Fusarium* species profile associated to the maize kernels analysed. However, at worldwide level, DON and ZEA occurrence in maize have been mostly associated to the contamination of *F. graminearum* and the related species of the *Fusarium sambucinum* clade [27], that are all DON and ZEA producing species [6,7,8,28]. In our results, *F. graminearum* was never isolated from the kernels analysed. This latter species is often associated to maize worldwide mostly in geographical areas where cold and wet environmental conditions characterise the period of maize cultivation. Since the majority of Iranian provinces sampled is characterised by hot and dry climate, the lack of *F. graminearum* and the predominance of *F. verticillioides* and *F. proliferatum* species is not surprising. 

Nevertheless, from our samples, we identified, based on the *EF-1α* gene sequences, with a significant frequency, also 32 strains belonging to FIESC, a complex of 31 phylogenetic species [29], and 3 strains of *F. brachygibbosum*, a closely related species. This is also reason for further concern, since FIESC isolates were shown to be potentially able to produce B-trichothecenes such as DON, due to the presence of *Tri5* gene within trichothecene biosynthetic loci [30]. Moreover, some members of this complex have been reported to be both ZEA and trichothecene producers [31,32], while a recent report by Villani et al. [33] reported the analysis of 13 FIESC members genomes, showing that the complete set of genes involved in the ZEA and trichothecene biosynthesis occurred in all genomes but also that the production of secondary metabolites, included the MTX, could be affected by the different distribution of functional and related gene clusters. The FIESC strains isolated in this study could produce in vitro both trichothecenes and ZEA, however their production was highly variable along the set of strains analysed, confirming the wide bio-diversity of FIESC reported in previous papers [31,32,33]. 

In our survey, two more species have been detected at a significant incidence: *F. thapsinum*, a member of FFSC, and the closely related species to FFSC *F. redolens*, formerly identified as *F. oxysporum* [34]. All isolates of both species produced enniatins and high level of BEA, which are structurally similar metabolites with demonstrated toxicity on human cell lines [26]. The occurrence of such metabolites cannot be neglected since, although they are not under any regulation by neither National nor International Institutions, they can contribute to increase the risk for human and animal health for the consumption of contaminated maize. 

Finally, it is interesting to notice that the profile of FBs produced in vitro by the strains of *F. verticillioides* and *F. proliferatum* isolated in our study was highly different between the two species, being both able to produce FB_1_, while production of FB_2_ and FB_3_ was detected only in *F. proliferatum* cultures. However the mean of FB_1_ production by *F. verticillioides* was much higher than *F. proliferatum*. These results show that, although no production of FB_2_ and FB_3_ was detected in vitro cultures of *F. verticillioides*, this species can be still considered as the main responsible for fumonisin accumulation in maize kernels in Iran. On the other hand, the population of *F. verticillioides* studied here needs to be further investigated at molecular level, in order to understand the genetic mechanism involved in the inhibition of FB_2_ and FB_3_ production. In addition, the extended occurrence on maize in Iran of *F. proliferatum* strains with full ability to produce in vitro all three more common and toxic FBs, shows that this species can be a very important contributor to the FB_1_ final contamination of maize kernels, being the *in vitro* production very useful in forecast contamination in the field.

## 4. Conclusions

In conclusion, our study showed an increase of *F. proliferatum* incidence in Iranian maize compared to previous reports and confirmed an extended concern for the potential contamination of maize kernels by FBs in the whole of Iran. Moreover, the occurrence of the FIESC species that are trichothecene and ZEA producers has been recorded for the first time in Iran, showing that a possible association of this complex to the ZEA contamination of maize kernels could occur. Finally, the diversity of *Fusarium* species isolated, compared the previous reports in Iran, highlights the possible increased risk for human and animal health because of the possible multi-toxin contamination of maize kernels in the whole country.

## 5. Materials and Methods

### 5.1. Sampling and Fungal Isolation

Maize samples of two crop seasons, 2015 and 2016, were collected in September–October from fields and maize grain silos of ten provinces of Iran: Khuzestan, Fars, Golestan, Ardabil, Alborz, Qazvin, Zanjan, Kermanshah, Lorestan and Isfahan (Figure 3). A total of 182 maize samples were collected in the main maize producing Iranian regions, as reported in Table 1. The sampling method was based on a hierarchical method [35]. Ears were put in paper boxes, labelled, moved to the laboratory, and air dried for 4–5 days in room temperature. 

Kernels were surface sterilized for 1 min in 3% sodium hypochlorite solution, rinsed twice in sterile distilled water, dried on filter paper and placed on Nash and Snyder medium containing 15 g peptone, 1 g K_2_HPO_4_, 0.5 g MgSO_4_·7H_2_O, 15 g agar, 1 g pentachloronitrobenzene (PCNB; Terraclor 75% WP), in 1 L distilled water. Petri dishes containing kernels were incubated at 25 °C in the dark for 5–7 days. All the fungal cultures developed from the kernels were transferred on potato dextrose agar (PDA) using a single-spore technique [36] and incubated at 25 °C for 7 days.

### 5.2. Morphological Identification

For morphological characterisation, pure cultures were transferred on both carnation leaf-piece agar (CLA; 20.0 g/L agar prepared in distilled water and piece of disinfected carnation leaf) and Synthetic Nutrient Agar (SNA; 1.0 g/L of KH_2_PO_4_, 1.0 g/L of KNO_3_, 0.5 g/L of MgSO_4_·7H_2_O, 0.5 g/L of KCl, 0.2 g/L of glucose, 0.2 g/L of sucrose and 20.0 g/L agar). Both CLA and SNA inoculated plates were incubated under the conditions described by Leslie and Summerell [36] for 2 weeks, to allow the fungal colonies growing. Macroscopic traits such as the colony appearance, colour, pigmentation and growth rate were observed on potato dextrose agar (PDA) according to Leslie and Summerell [36]. Morphological identification was performed based on the morphological characteristics observed at optical microscope as described by Booth [37], Gerlach and Nirenberg [38], and Leslie and Summerell [36].

### 5.3. Molecular Identification

For molecular identification, 234 representative isolates from different morphologically identified *Fusarium* species, were selected and grown on PDA medium. Mycelium was lyophilized for total DNA extraction, and then 10 mg of lyophilized mycelium, grinded with 5 mm iron beads, was processed with “Wizard^®^ Magnetic DNA Purification System for Food” kit (Promega, Fitchburg, WI, USA). The quality of genomic DNA was determined by agarose gel electrophoresis and the quantification by using the Spectrophotometer ND-1000 (NanoDrop, Thermo-Scientific, Waltham, MA, USA). Molecular identification was carried out based on elongation factor (*EF-1α*) gene sequencing. The region between primers EF1 and EF2 from O’Donnell et al. [39] was amplified. The reaction mixture for polymerase chain reaction (PCR) was prepared in a final volume of 15 μL, consisting of DNA (1.5 μL), buffer 1X, primers 300 nM, dNTPs 200 nM, Taq DNA polymerase 0.5 U. The PCR conditions were as follows: 95 °C for 2 min; 35 cycles of 95 °C for 30 s, 60 °C for 40 s and 72 °C for 50 s. Final extension was performed at 72 °C for 7 min. The PCR amplification products were verified on 1.5% agarose gel.

For sequencing, PCR products were purified with the enzymatic mixture EXO/FastAP (Exonuclease I, FastAP thermosensitive alkaline phosphatase, Thermo Scientific, Waltham, MA, USA). BigDye Terminator v3.1 Cycle Sequencing Ready Reaction Kit (Thermo Scientific, Waltham, MA, USA) was used for sequence reactions of both strands, and then the PCR products were purified by gel filtration through Sephadex G-50 (5%) (Sigma Aldrich, Milan, Italy) and run on the 3730xl DNA Analyzer (Applied Biosystems, Foster City, CA, USA). DNA sequences were analysed with the Sequencing Analysis 5.2 software (Applied Biosystems, Waltham, MA, USA). Each consensus sequence was generated from forward and reverse strand with Bionumerics software (Applied Maths, Kortrijk, Belgium). To obtain a previous species identification in order to select reference sequences for *Fusarium* species to be used in the phylogenetic analysis, the EF-1α sequences were searched on GeneBank database by using the Basic Local Alignment Tool (BLAST). In addition to the EF-1α sequences of 234 isolates from Iranian maize, 39 reference sequences for *Fusarium* species and one *Trichoderma* sequence used as outgroup, were downloaded from GeneBank and used for phylogenetic analysis. All the sequences were aligned by using the MUSCLE algorithm [40] with MEGA7 software ver. 7.0.14 [41]. The evolutionary history was inferred by using the Maximum Likelihood method based on the Tamura–Nei model [42] in MEGA 7 software. To evaluate the support for inferred topologies, the percentage of trees in which the associated taxa clustered together was evaluated by using bootstrapping [43] with 1000 replicates.

### 5.4. Mycotoxins Analysis

Selected 104 *Fusarium* isolates, representative of the *Fusarium* species belonging to *F. fujikuroi* species complex (FFSC) and related species (*F. redolens*), were examined for mycotoxin production. In particular, 68 isolates of *F. verticillioides*, 26 *F. proliferatum*, 3 *F. subglutinans*,1 *F. temperatum*, 1 *F. nygamai* and 5 *F. redolens* were analysed.

For mycotoxin production assays, we used 30 g rice in PYREX Glass Erlenmeyer Flasks, added with 13.5 mL of distilled water, standing overnight, and then autoclaved at 121 °C for 30 min. The flasks containing autoclaved rice were inoculated with piece of fungal cultures grown on PDA and incubated at 25 °C for 21 days in order to allow fungal development and mycotoxin production. High-performance liquid chromatography (HPLC) analytical methods were used to detect fumonisins B_1_, B_2_ and B_3_ (FB_1_, FB_2_, FB_3_), beauvericin (BEA), enniatins (ENNs) A, A_1_, B and B_1_.

#### 5.4.1. Fumonisin Production

Fumonisin production was analysed for *F. verticillioides* (68 isolates), *F. proliferatum* (26 isolates) and *F. nygamai* (1 isolate). One gram of inoculated rice culture was used for toxin extraction with 5 mL of methanol/water (75/25, *v/v*). Samples were placed for 60 min in an orbital shaker, and then filtered using Whatman no. 4 filter papers (Maidstone, UK). 500 µL was diluted with 500 µL ultrapure water (Millipore, Bedford, MA, USA). An aliquot of 50 µL of the extract was derivatized with 50 µL of o-phtaldialdehyde (OPA) using the HPLC autosampler Agilent 1100 (Agilent, Waldbronn, Germany). The column Symmetry shield RP18 15 cm × 4.6 mm, 5 µm (Waters, Milford, MA, USA) with a guard column inlet filter (0.5 µm × 3 mm diameter, Rheodyne Inc. Rohnert Park, CA, USA) was thermostat set at 30 °C, and FLD detector was set at ex = 335 nm, em = 440 nm. A volume of 100 µL was injected in the HPLC at 3 min after adding the OPA reagent. The mobile phase consisting of a binary gradient was applied as follows: the initial composition of the mobile phase 57% of A water-acetic acid (99/1, *v/v*)/43% of B acetonitrile-acetic acid (99/1, *v/v*) was kept constant for 5 min, then B solvent was linearly increased to 54% in 21 min, then up to 58% at 25 min and kept constant for 5 min. The flow rate of the mobile phase was 0.8 mL/min. The retention time of the FB_1_ was about 16.6 min, 24.6 min for FB_2_ and 26.0 min for FB_3_. The mycotoxins were quantified by comparing peak areas with a calibration curves obtained with standard solutions (Romer Labs Diagnostic GmbH, Tulln, Austria). The detection limits (LOD) of the method based on signal-to-noise ratio of 3:1 was 12.5 µg/kg for FB_1_, FB_2_ and FB_3_.

#### 5.4.2. Beauvericin and Enniatins Production

Production of BEA and ENNs was assayed for *F. redolens* (5 isolates), *F. subglutinans* (3 isolates) and *F. temperatum* (1 isolate). Fungal culture on rice (1 g) was used for toxin extraction with 5 mL of methanol/water (70/30, *v/v*) on an orbital shaker for 60 min, and then was filtered using Whatman no. 4 filter papers (Waters, Milford, MA, USA). The sample (100 µL) was diluted with 900 µL ultrapure water (Millipore, Bedford, MA, USA) and filtered using RC through 0.20 µm regenerated cellulose filter (Phenomenex, Torrance, CA, USA). A volume of 100 µL was injected into HPLC apparatus (Agilent 1260 Series, Agilent Technology, Santa Clara, CA, USA). The analytical column was a Gemini (150 × 4.6 mm, 5 μm, Phenomenex) preceded by a pre-column Gemini (4 × 3 mm, Phenomenex). The mobile phase consisting of a binary gradient was applied as follows: the initial composition of the mobile phase 30% of A water/70% of B acetonitrile was kept constant for 5 min, then B solvent was lineary increased to 90% in 10 min. The flow rate of the mobile phase was 1 mL/min. Retention time was 11.4 min for BEA, 9 min for ENN B, 10.3 min for ENN B_1_, 12 min for ENN A_1_ and 13 min for ENN A. The mycotoxins were quantified by comparing peak areas with a calibration curves obtained with standard solutions (Sigma-Aldrich, Milan, Italy). The detection limits (LOD) of the method based on signal-to-noise ratio of 3:1, were the following: BEA = 10 µg/kg, ENN B = 60 µg/kg, ENN B_1_ = 70 µg/kg, ENN A = 200 µg/kg, ENN A_1_ = 500 µg/kg.

## Figures and Tables

**Figure 1 toxins-11-00297-f001:**
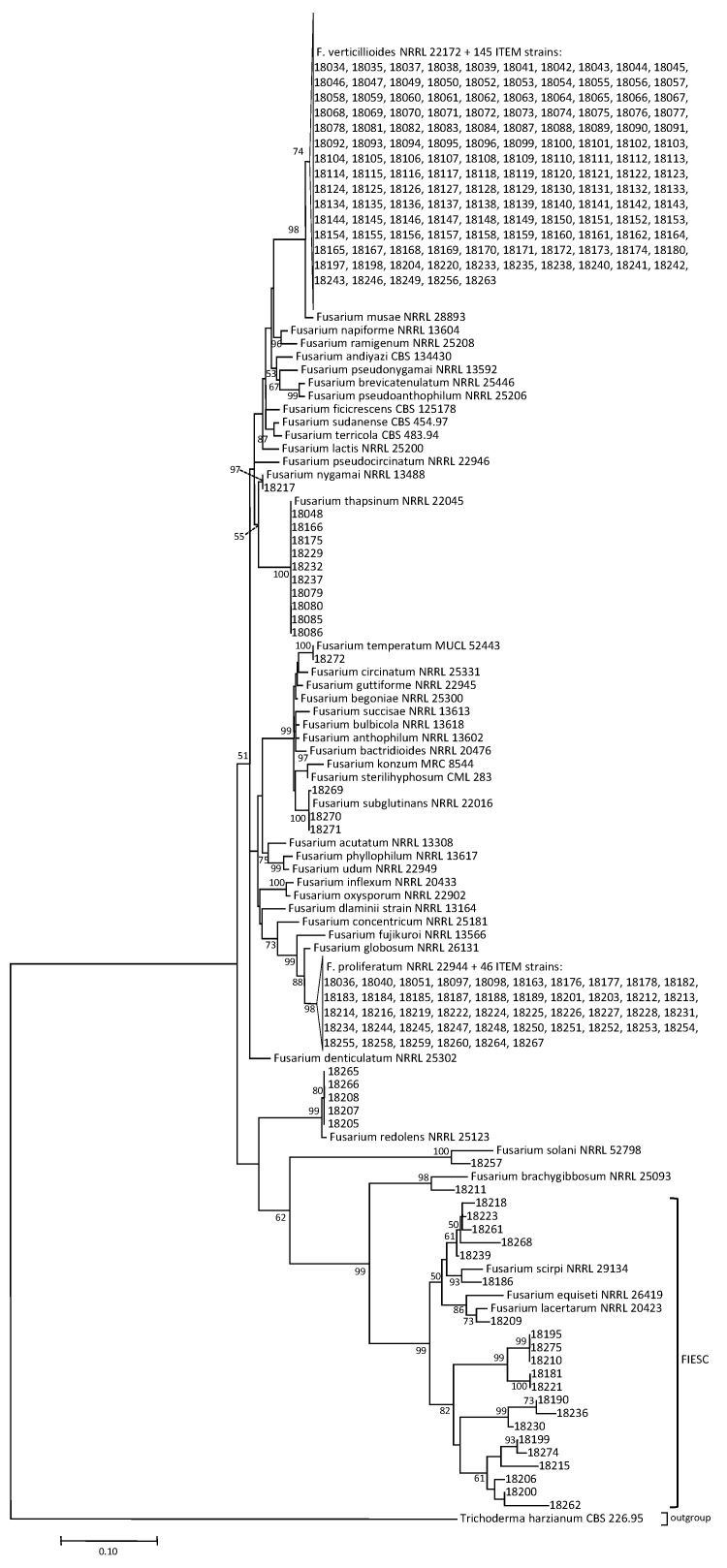
Phylogenetic tree based on *EF-1α* gene sequences, inferred by using the Maximum Likelihood method, on 234 *Fusarium* isolated from Iranian maize, compared to reference sequences for *Fusarium* species. The bootstrap values are shown next to the branches.

**Figure 2 toxins-11-00297-f002:**
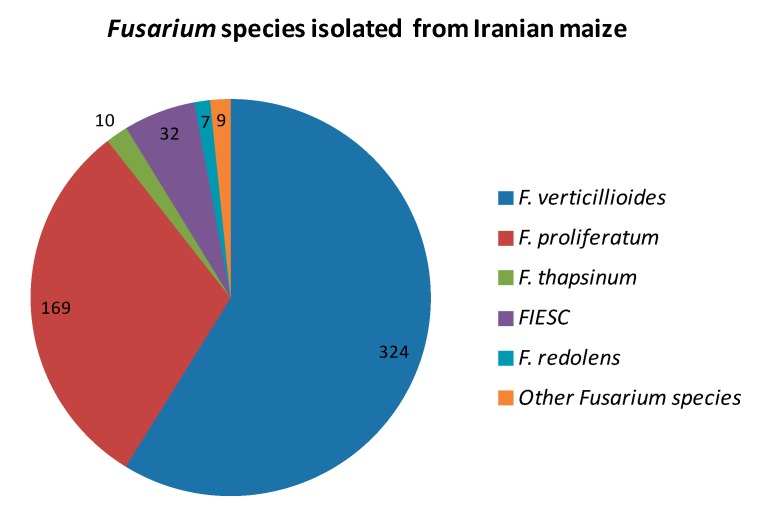
Distribution of *Fusarium* species isolated from 182 maize samples collected in 10 maize producing provinces of Iran. The number of isolates for each species is reported.

**Figure 3 toxins-11-00297-f003:**
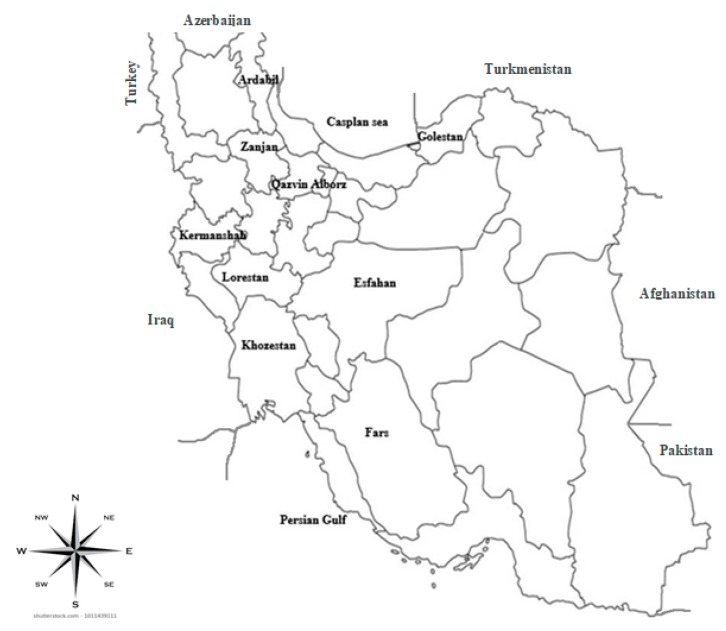
Map depicting the location of ten provinces of Iran where maize kernel samples were harvested in two maize growing seasons, 2015 and 2016.

**Table 1 toxins-11-00297-t001:** Number of *Fusarium* isolates from maize kernels collected in 10 different provinces of Iran. The occurrence and the incidence of the most occurring species *F. verticillioides* (Fv) and *F. proliferatum* (Fp) are also reported.

Iranian Provinces	Number of Maize Sample	Number of *Fusarium* Isolates	*Fusarium* Species Occurrence (%)			Incidence * (%)		
			Fv	Fp	Other Species	*Fusarium* spp.	Fv	Fp
Alborz	30	126	47	48	5	21	10	10
Golestan	15	100	61	28	11	33	20	9
Qazvin	25	84	56	40	4	17	9	7
Fars	30	82	50	34	16	14	7	5
Khuzestan	24	62	74	0	26	13	10	0
Ardabil	20	43	81	12	7	11	9	1
Zanjan	12	27	78	18	4	11	9	2
Lorestan	10	11	18	55	27	6	1	3
Esfahan	10	9	78	0	22	5	3	0
Kermanshah	6	7	71	28	0	6	4	2
TOTAL	182	551	59	31	10	15	9	5

* Incidence was calculated as percentage of seeds infected on total analysed seeds.

**Table 2 toxins-11-00297-t002:** Fumonisin B_1_ (FB_1_), B_2_ (FB_2_) and B_3_ (FB_3_) production, expressed in µg/g, of *F. verticillioides* and *F. proliferatum* strains from Iranian maize kernels. For each mycotoxin, the number of positive strains on total analysed strains, and range and mean of production with standard error (SE) are reported.

Fumonisin Producing Strains	*F. verticillioides ***	*F. proliferatum ***			
	FB_1_ *	FB_1_	FB_2_	FB_3_	Total FBs
N. positive strains/total	67/67	26/26	17/26	11/26	26/26
Range (µg/g)	79–2232	1–1860	0–466	0–11	1–2335
Mean ± SE (µg/g)	505 ± 38	216 ± 69	30 ± 18	3 ± 1	238 ± 442

* FB_2_ and FB_3_ were not detected in any *F. verticillioides* culture. ** For each mycotoxin, the number of positive strains on total analysed strains, and range and mean of production are reported.

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
