# Peer review of "Isolation, Molecular Identification and Mycotoxin Profile of Fusarium Species Isolated from Maize Kernels in Iran"

_toxins, 2019, doi:10.3390/toxins11050297_

Round 1

Reviewer 1 Report

There is no abstract. Please add. 

Please explain the FIESC and FSSC abbreviations on first mentioning, not all the way in the line 87. And please explain the meaning of these complexes. 

Why did you use rice for mycotoxin production? Why not maize?

Table 2 is very confusing. Please corrrect it and make it more reader-friendly. I do not understand what are the values stated in in: ie. 26/26... What does it stand for?  Why did not you measure all other FBs for F. verticillioides, or better to say, please note that they were not detected in that sample. 

Author Response

There is no abstract. Please add.

Done. It was a past and copy mistake 

Please explain the FIESC and FSSC abbreviations on first mentioning, not all the way in the line 87. And please explain the meaning of these complexes. 

We reported the acronym of the two species complex the first time we cited in the paper

Why did you use rice for mycotoxin production? Why not maize?

The rice was selected since triggers much more than maize the in vitro production of mycotoxins by the Fusarium species. Moreover, the maize is constantly contaminated by fumonisins, therefore we wanted to use a substrate surely fumonisins free.

Table 2 is very confusing. Please corrrect it and make it more reader-friendly. I do not understand what are the values stated in in: ie. 26/26... What does it stand for?  Why did not you measure all other FBs for F. verticillioides, or better to say, please note that they were not detected in that sample. 

We reported under the table the following sentence already reported in the title of the table: *FB2 and FB3 were not detected in any F. verticillioides culture. ** For each mycotoxin, the number of positive strains on total analysed strains, and range and mean of production are reported.

All FBs production were investigated also for F. verticillioides. No strain produced the FB2 and FB3. We have discussed it in the paper in lines 314-322

Reviewer 2 Report

This manuscript reported an extensive collection of Fusarium species from maize kernels in Iran, and performed identification of Fusarium species using morphological characteristics as well as EF1a gene sequence phylogeny. The authors also used HPLC to analyze the mycotoxin contents produced by certain Fusarium species. In general, this is a complete, unique and timely report about a nationwide survey in Iran. I believe this report would draw a lot of interest from readers of the Toxins journal.

In general, most parts of this manuscript should be fine after moderate language polishing. The authors did a good job in the discussion part to relate their findings to previous reports, focusing on those in Iran. But the first paragraph is too long and can be divided into several shorter paragraphs to make it easier to read. However, I suggest the authors to spend more time re-write the introduction. The authors rushed into introducing toxins too quickly, and there should be a bit more words about the fungal biology of Fusarium species. Also, it would be very helpful to lay out a general background of epidemiology of fusarium caused diseases in Iran.

Some detailed questions:

Result 2.1. It would be nice to show photos (maybe as supplemental files?) to present how you determined Fusdraium spp with morphological characteristics. It is fine if you do not.

Result 2.1. Table 1. Please define “incidence (%)”.

Result 2.1. Need to provide primers sequences for amplifying EF1alpha gene.

Result 2.1. For molecular identification of Fusarium species, using species-specific PCR reactions are more reliable than constructing a phylogenetic tree using one gene’s sequence. For F. verticillioides and F. proliferatum, there are available species-specific PCR reactions, I suggest to test a few isolates identified as Fv or Fp species in this study using the Fv and Fp species-specific PCR reactions to confirm the genotype.

Figure 2 showing distribution of Fusarium species seems unnecessary.

Result 2.2. mycotoxin production, table 2, better show separate data points for each and every Fv and Fp isolate, instead of just showing a mean value. Because different isolates of the same species could vary a lot. And when showing the mean value, it’s better to show as mean ± SD or SE.

Result 2.2. Table 2. FB2 and FB3 were not detected in any Fv samples, why? I have an impression from the discussion that this was likely due to the method of using in vitro rice culture for toxin production? But other similar studies use the same method and they can identify FB2 and FB3 from Fv species, right?

Formatting issues:

Abstract is missing from the PDF version.

Chemical names subscript form (eg. KH2PO4 in L260, etc.)

Author Response

Reviewer 2

This manuscript reported an extensive collection of Fusarium species from maize kernels in Iran, and performed identification of Fusarium species using morphological characteristics as well as EF1a gene sequence phylogeny. The authors also used HPLC to analyze the mycotoxin contents produced by certain Fusarium species. In general, this is a complete, unique and timely report about a nationwide survey in Iran. I believe this report would draw a lot of interest from readers of the Toxins journal.

 In general, most parts of this manuscript should be fine after moderate language polishing. The authors did a good job in the discussion part to relate their findings to previous reports, focusing on those in Iran. But the first paragraph is too long and can be divided into several shorter paragraphs to make it easier to read.

Done

However, I suggest the authors to spend more time re-write the introduction. The authors rushed into introducing toxins too quickly, and there should be a bit more words about the fungal biology of Fusarium species. Also, it would be very helpful to lay out a general background of epidemiology of fusarium caused diseases in Iran.

Done: lines 87-93

Some detailed questions:

Result 2.1. It would be nice to show photos (maybe as supplemental files?) to present how you determined Fusdraium spp with morphological characteristics. It is fine if you do not.

Unfortunately, we did not make pictures of the Fusarium strains analyzed morphologically, but we used the taxonomic key reported in Leslie and Summerell book on Fusarium

Result 2.1. Table 1. Please define “incidence (%)”.

Reported the definition under Table 1

Result 2.1. Need to provide primers sequences for amplifying EF1alpha gene.

Since we did not develop these primers, we cited at paragraph 5.3. the reference number 39 of O’Donnell et al., where the primer sequences are reported

Result 2.1. For molecular identification of Fusarium species, using species-specific PCR reactions are more reliable than constructing a phylogenetic tree using one gene’s sequence. For F. verticillioides and F. proliferatum, there are available species-specific PCR reactions, I suggest to test a few isolates identified as Fv or Fp species in this study using the Fv and Fp species-specific PCR reactions to confirm the genotype.

We have developed at ISPA our own primers for FV and FP species (Mulè et al., 2004. European Journal of Plant Pathology), and we have used extensively for species confirmation. We did also for this set of FV and FP for our internal confirmation of good primers quality, once we identified by EF sequences the strains. This gene is considered by all Fusarium expertises as the best one to identify the different Fusarium species and we preferred to built a phylogenetic tree in order to show the genetic diversity of all species isolated from maize kernels in this survey

Figure 2 showing distribution of Fusarium species seems unnecessary.

Sorry, we would prefer to keep this figure to provide a direct visual impact of the quantitative relationships among the species.

Result 2.2. mycotoxin production, table 2, better show separate data points for each and every Fv and Fp isolate, instead of just showing a mean value. Because different isolates of the same species could vary a lot. And when showing the mean value, it’s better to show as mean ± SD or SE.

We analyzed chemically a total of 93 strains of FV and FP, so the table would be too much long and not so meaningful. We would prefer to present the data leaving this table. However, we have added the SE

Result 2.2. Table 2. FB2 and FB3 were not detected in any Fv samples, why? I have an impression from the discussion that this was likely due to the method of using in vitro rice culture for toxin production? But other similar studies use the same method and they can identify FB2 and FB3 from Fv species, right?

The rice was selected as substrate since triggers much more than maize the in vitro production of mycotoxins by the Fusarium species. Moreover, the maize is constantly contaminated by fumonisins, therefore we wanted to use a substrate surely fumonisins free. All FBs production were investigated also for F. verticillioides. No strain produced the FB2 and FB3. We have discussed it in the paper in lines 314-322. However, the F. proliferatum strains could produce them, therefore is not a problem of substrate. We are investigating the reason of such lack of production by only the FV strains.

Formatting issues:

Abstract is missing from the PDF version.

Done. It was a past and copy mistake

Chemical names subscript form (eg. KH2PO4 in L260, etc.)

Done

Round 2

Reviewer 2 Report

I am glad to see that the authors have properly addressed most of my comments and suggestions during the first round of revision. Still, I think there are two points that the authors need to do to improve the manuscript:

For table 2, as I mentioned in previous comments, it is necessary to present detailed information of every strain in addition to a summarized table, especially considering the large variation of FBs measured between different strains (based on the range and SE). I understand that there are 93 strains measured, and it is not likely and may not be necessary to show all of them in the main text, but please submit such data sheet as supplemental files.

Still, language, typo error and formatting check needed, particularly in the reference list.

Author Response

Thank you for your further suggestions and comments. We followed the reviewer indications as shown below:

For table 2, as I mentioned in previous comments, it is necessary to present detailed information of every strain in addition to a summarized table, especially considering the large variation of FBs measured between different strains (based on the range and SE). I understand that there are 93 strains measured, and it is not likely and may not be necessary to show all of them in the main text, but please submit such data sheet as supplemental files.

We add the Table as Supplemental file

Still, language, typo error and formatting check needed, particularly in the reference list.

We tried to edit and correct all tupe errors detected.